# Ferricyanide photo-aquation pathway revealed by combined femtosecond Kβ main line and valence-to-core x-ray emission spectroscopy

Marco Reinhard [1]✉, Alessandro Gallo [1], Meiyuan Guo [1], Angel T. Garcia-Esparza [1], Elisa Biasin [2], Muhammad Qureshi[1], Alexander Britz[1], Kathryn Ledbetter [3,5], Kristjan Kunnus[1], Clemens Weninger[1,6], Tim van Driel[1], Joseph Robinson[1], James M. Glownia[1], Kelly J. Gaffney [1], Thomas Kroll[1], Tsu-Chien Weng[4], Roberto Alonso-Mori [1] ✉ & Dimosthenis Sokaras [1]✉

Reliably identifying short-lived chemical reaction intermediates is crucial to elucidate reaction mechanisms but becomes particularly challenging when multiple transient species occur simultaneously. Here, we report a femtosecond x-ray emission spectroscopy and scattering study of the aqueous ferricyanide photochemistry, utilizing the combined Fe Kβ main and valence-to-core emission lines. Following UV-excitation, we observe a ligand-to-metal charge transfer excited state that decays within 0.5 ps. On this timescale, we also detect a hitherto unobserved short-lived species that we assign to a ferric penta-coordinate intermediate of the photo-aquation reaction. We provide evidence that bond photolysis occurs from reactive metal-centered excited states that are populated through relaxation of the charge transfer excited state. Beyond illuminating the elusive ferricyanide photochemistry, these results show how current limitations of Kβ main line analysis in assigning ultrafast reaction intermediates can be circumvented by simultaneously using the valence-to-core spectral range.

Light induced charge transfer (CT) processes in transition metal complexes play a crucial role in light-harvesting in natural and synthetic materials[1–5]. Because the photocatalytic efficiency of such systems depends critically on the lifetime of the photoexcited CT excited states (ES), intense efforts have been directed towards elucidating the key factors that allow extending CT ES lifetimes of cost-effective, non-toxic Fe-based light-harvesters[6–8]. Often however, the CT ES of these complexes are rapidly deactivated via non-radiative processes that involve potential energy surface crossings with metal-centered (MC) ES. The population of MC ES in turn, may enhance the propensity of a compound towards irreversible decomposition via rapid metal-ligand bond dissociation and the subsequent formation of coordinatively unsaturated, reactive intermediates and long-lived photoproducts. Bond photolysis can also occur when populating CT ES, but the responsible mechanisms remain less well understood[9,10]. Numerous time-resolved ultrafast studies have therefore addressed the

[1]SLAC National Accelerator Laboratory, Menlo Park, CA, USA. [2]Physical Sciences Division, Pacific Northwest National Laboratory, Richland, WA, USA. [3]Department of Physics, Stanford University, Stanford, CA, USA. [4]School of Physical Science and Technology, ShanghaiTech University, Shanghai, China. [5]Present address: Department of Physics, Harvard University, Cambridge, MA, USA. [6]Present address: MAX IV Laboratory, Lund University, Lund, Sweden. ✉e-mail: marcor@slac.stanford.edu; robertoa@slac.stanford.edu; dsokaras@slac.stanford.edu

deactivation mechanisms of CT ES in Fe-based complexes as a function of ligand architecture and solvent environment[11]. Here, we focus on the aqueous ferricyanide anion $^{2S+1}[Fe^{III}(CN)_6]^{3-}$ which has a total spin quantum number S = ½ in the electronic ground state.$^2[Fe^{III}(CN)_6]^{3-}$ is an archetypal model compound whose photochemistry poses some long standing questions[12–15]. The UV-visible absorption spectrum (Fig. 1a) exhibits a combination of ligand-to-metal charge transfer (LMCT) and MC transitions[15,16]. Recent ultrafast studies have utilized ~400 nm excitation of solvated $^2[Fe^{III}(CN)_6]^{3-}$ into the $^2T_{1u}$ LMCT ES ($t_{1u} \rightarrow t_{2g}$), however reaching different conclusions about the ES lifetimes and ground state recovery pathways[17–20]. LMCT ES lifetimes extending well beyond a picosecond in acetonitrile and dimethyl sulfoxide solutions were reported by Zhang et al.[20], using femtosecond infrared transient absorption spectroscopy, but subsequent studies demonstrate the longer-lived transient features observed in their cyanide stretch absorption spectrum reflect a vibrationally hot electronic ground state, rather than longer-lived electronic ES dynamics[17,21]. Using photoelectron spectroscopy, Engel et al.[18] and Ojeda et al.[17] both concluded that the photoexcited $^2T_{1u}$ LMCT ES decays on a sub-picosecond timescale in water. Based on their observed decay kinetics and density functional theory calculations, Engel et al. also invoked a Jahn-Teller distorted quartet MC ES populated within ~170 fs from the decaying $^2T_{1u}$ LMCT ES. This quartet MC ES then relaxes back into the $^2T_{2g}$ electronic ground state within ~780 fs. In contrast, Ojeda et al. found a somewhat slower $^2T_{1u}$ LMCT ES lifetime of ~0.5 ps, however they did not observe evidence of a quartet MC ES nor provide an alternative mechanistic explanation for the observed rapid, sub-picosecond ground state recovery.

For the longer-lived photoproducts of aqueous $^2[Fe^{III}(CN)_6]^{3-}$, the optical wavelength dependence, low quantum yields (2 – 6%) and reaction mechanisms[12] are also poorly understood, and the role of the MC ES in the $^2[Fe^{III}(CN)_6]^{3-}$ photochemistry remains particularly

elusive. The presence of the ferric, aquated complex ($^2[Fe^{III}(CN)_5H_2O]^{2-}$) resulting from the photoinduced substitution of a cyanide anion by a water molecule was reported by Fuller et al.[12]:

$$^2[Fe^{III}(CN)_6]^{3-} \xrightarrow{h\nu} {}^2[Fe^{III}(CN)_6]^{3-*} \xrightarrow{+H_2O} {}^2[Fe^{III}(CN)_5H_2O]^{2-} + CN^- \quad (1)$$

In contrast, Ojeda et al. have assigned a long-lived component in their data to a small persistent fraction of the ferrous, aquated complex ($^1[Fe^{II}(CN)_5H_2O]^{3-}$)[17]. Moreover, they have proposed that this species is produced in low yields via a thermally activated pathway, directly from the photoexcited $^2T_{1u}$ LMCT ES but they did not have evidence of a penta-coordinate reaction intermediate.

Femtosecond Fe Kβ main line (3p-1s) x-ray emission spectroscopy (XES) has been successfully utilized to identify short-lived CT and MC ES with different spin multiplicities in solvated transition metal complexes[22–25]. In some cases, however, complementary information is needed because the sensitivity of Fe Kβ main line XES to metal-ligand covalency[26,27] and more subtle ligand field effects[28] can complicate the assignment of spectra to distinct species. The simultaneous recording of x-ray solution scattering (XSS) is experimentally a convenient extension (Fig. 1b) but the global ensemble nature combined with the limited information content of azimuthally integrated scattering curves often requires extensive and hypothesis-based modeling to extract reliable dynamic structural information[25,29,30]. The valence-to-core (VtC) x-ray emission lines have a pronounced sensitivity to metal-ligand bonding and structure[31] hence being an attractive probe of the photoinduced dynamics that may circumvent the limitations of the Fe Kβ main lines. Importantly, VtC emission lines can be accurately simulated with density functional theory (DFT) eliminating the need of measuring reliable candidate reference spectra. However, VtC lines exhibit a 20–100 times lower intensity compared to Kβ main lines (Fig. 1c) and that has been a significant barrier for efficiently engaging

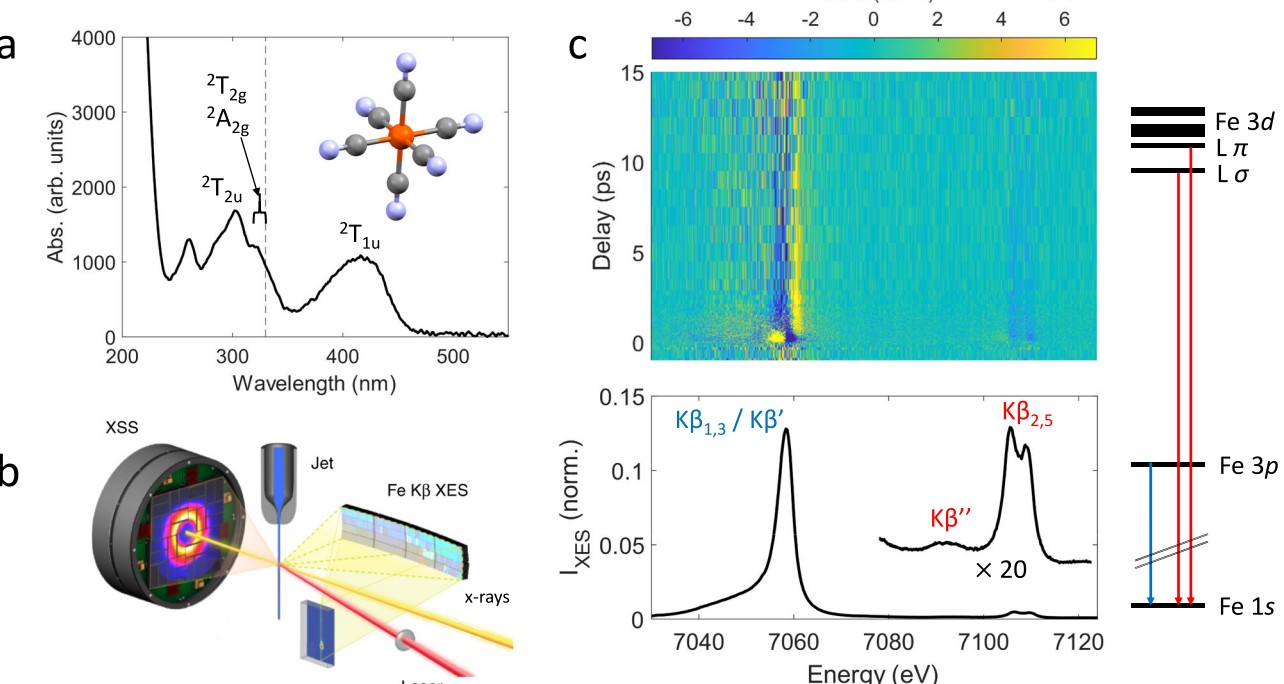

**Fig. 1 | Photoinduced dynamics of aqueous ferricyanide observed with femtosecond x-ray emission spectroscopy and scattering. a** UV-visible absorption spectrum of aqueous $^2[Fe^{III}(CN)_6]^{3-}$. Electronic excited states that are relevant for this work are shown. The photoexcitation wavelength (336 nm) is represented by the black dashed line. **b** Experimental setup used to collect simultaneous femtosecond Fe Kβ main line (Kβ$_{1,3}$/Kβ') and valence-to-core (Kβ$_{2,5}$/Kβ'') x-ray emission spectroscopy and solution scattering at the X-ray Correlation Spectroscopy instrument of the Linac Coherent Light Source (adapted from Kjaer et al.[29] with permission from the Royal Society of Chemistry). **c** The time dependent difference between pumped and unpumped x-ray emission spectra (top) is shown together with the $^2[Fe^{III}(CN)_6]^{3-}$ ground state spectrum (bottom). Source data are provided as a Source Data file.

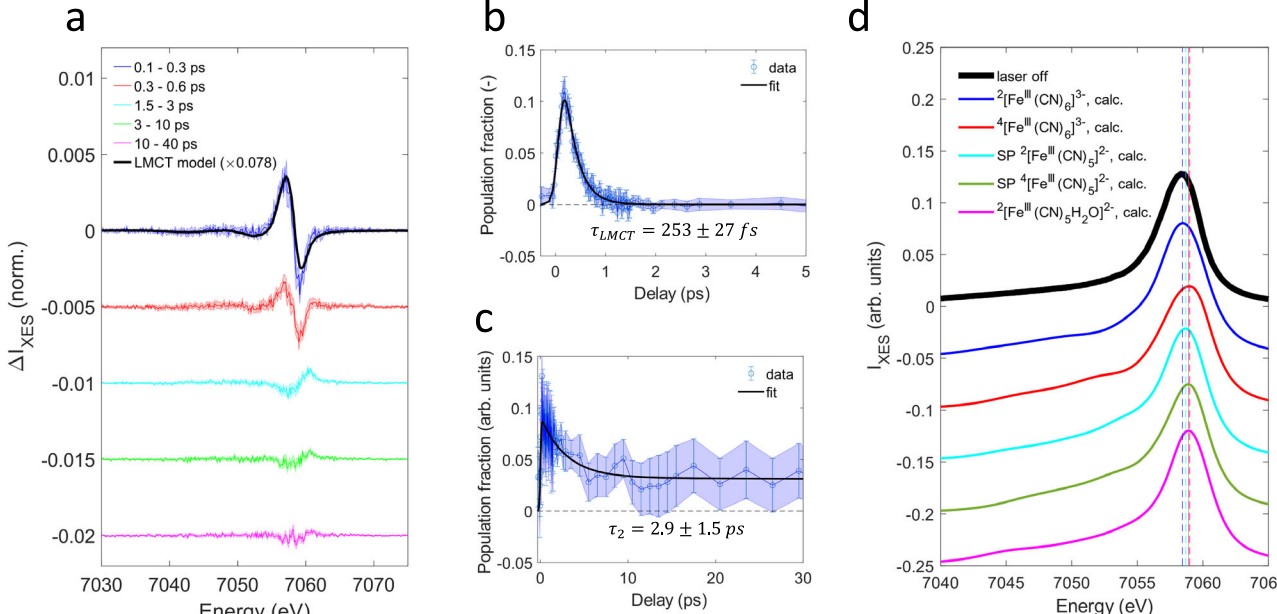

**Fig. 2 | Fe Kβ main line x-ray emission. a** Fe Kβ main line difference spectra averaged in different time bins. Shaded areas reflect the standard deviation within a time bin when all difference spectra are rescaled to the bin-averaged summed difference signal magnitude. The black line (LMCT model) represents the ground state difference of aqueous $^1$[Fe$^{II}$(CN)$_6$]$^{4-}$ and $^2$[Fe$^{III}$(CN)$_6$]$^{3-}$. **b** Ligand-to-metal charge transfer excited state population ($N_{LMCT}$) extracted from the fit of the main line region of the difference map shown in Fig. 1c. The uncertainty at each time delay was estimated using a cutoff in the increase of the sum of squared residuals when varying the population fraction with respect to the optimized value.
**c** Population of the longer-lived species ($N_{S2}$) extracted from the fit of the main line region of the difference map shown in Fig. 1c. The uncertainty at each time delay is estimated using a cutoff in the increase of the sum of squared residuals when varying the population fraction with respect to the optimized value. **d** The experimental $^2$[Fe$^{III}$(CN)$_6$]$^{3-}$ ground state spectrum (black) is shown together with calculated spectra for $^2$[Fe$^{III}$(CN)$_6$]$^{3-}$, $^4$[Fe$^{III}$(CN)$_6$]$^{3-}$, square pyramidal (SP) $^2$[Fe$^{III}$(CN)$_5$]$^{2-}$, SP $^4$[Fe$^{III}$(CN)$_5$]$^{2-}$ and $^2$[Fe$^{III}$(CN)$_5$H$_2$O]$^{2-}$. All calculated spectra were aligned to their center of mass, then shifted by 1.1 eV to overlap the experimental and calculated $^2$[Fe$^{III}$(CN)$_6$]$^{3-}$ Kβ$_{1,3}$ peaks. Peak positions of the calculated spectra are indicated by the vertical dashed lines. The peak positions of SP $^4$[Fe$^{III}$(CN)$_5$]$^{2-}$ and $^2$[Fe$^{III}$(CN)$_5$H$_2$O]$^{2-}$ overlap.

the VtC region in ultrafast time-resolved studies. Emerging capabilities at x-ray free electron laser sources and high throughput x-ray spectrometers now enable the utilization of the VtC spectral range on the femtosecond timescales[32].

Here, we leverage these novel ultrafast x-ray spectroscopy developments by combining femtosecond Fe Kβ main line and VtC XES (Fig. 1b) to further explore the mechanistic details of the LMCT ES deactivation and the generation of long-lived photochemical reaction products in aqueous $^2$[Fe$^{III}$(CN)$_6$]$^{3-}$. We utilize 336 nm excitation, close to the $^2$T$_{2u}$ LMCT ES ($t_{2u} \rightarrow t_{2g}$) reported around ~300 nm and two MC ES ($^2$A$_{2g}$ and $^2$T$_{2g}$) reported in the 309–336 nm range[15,16]. Importantly, these MC ES were inaccessible in previous femtosecond studies using 400 nm excitation but may have been accessed in a time-resolved x-ray absorption study with ~70 ps resolution that found evidence for different long-lived photoproducts using 355 nm and 266 nm excitation while being unable to unambiguously assign them[33]. Both $^2$A$_{2g}$ and $^2$T$_{2g}$ MC ES arise from the HOMO-LUMO electronic transition ($t_{2g} \rightarrow e_g$) that promotes an electron into an $e_g$-orbital that is antibonding with respect to the Fe-CN bonds, thus potentially providing a rationale for cyanide labilization[33,34]. By using the combined Fe Kβ main line and VtC spectral range of photoexcited aqueous $^2$[Fe$^{III}$(CN)$_6$]$^{3-}$, we unambiguously determine the LMCT ES lifetime, and detect a hitherto unobserved transient species associated with the photo-aquation process.

## Results

### Fe Kβ main line x-ray emission spectra

The combined Kβ main line and VtC ground state and difference signals (pumped minus unpumped) are shown in Fig. 1c as a function of the pump probe delay. In this section we focus on the Kβ main line region that comprises the stronger Kβ$_{1,3}$ peak around ~7058 eV and the

weaker Kβ′ feature around ~7045 eV. Within the first ~300 fs, the Kβ main line difference spectrum indicates a shift of the pumped spectrum towards lower emission energies (Fig. 2a). This signal decays within a picosecond, revealing a longer-lived difference signal that reflects a shift towards higher energies. This longer-lived component partially decays within 1–5 ps, leaving a small residual that persists up to 40 ps, i.e., the upper time delay limit of our measurement. While subtle spectral reshaping may occur in the 1.5–40 ps range, these changes are not unambiguously resolved (Supplementary Note 1, Supplementary Fig. 1).

We first discuss the short-lived spectral features that decay within the first few hundred femtoseconds. Near the pump wavelength of 336 nm, the UV-visible absorption spectrum (Fig. 1a) exhibits both LMCT and MC transitions. The symmetry-allowed $^2$T$_{2u}$ LMCT ES is centered near 300 nm and the symmetry-forbidden $^2$A$_{2g}$ and $^2$T$_{2g}$ MC ES were assigned in the 309 − 336 nm range[15,16]. We may therefore populate a mixture of the LMCT and MC ES. The observed rapid Kβ main line shift towards lower emission energies indicates a transient reduction of the Fe(III) center and formation of a singlet state at the metal site. As shown in Fig. 2a, a comparison between difference spectra in the 100 − 300 fs range and an LMCT model difference spectrum indeed shows good agreement, thus confirming that the dominant contribution on this timescale stems from the formation of an LMCT ES. Small deviations indicate the early presence of the second species that persists at longer time delays (Supplementary Note 2, Supplementary Fig. 2). The LMCT model difference was constructed from the $^1$[Fe$^{II}$(CN)$_6$]$^{4-}$ and $^2$[Fe$^{III}$(CN)$_6$]$^{3-}$ ground state spectra collected during the same experiment as the transient data, but in the absence of the pump laser. While both the $^1$[Fe$^{II}$(CN)$_6$]$^{4-}$ ground state and the LMCT ES exhibit a fully occupied $t_{2g}$-subshell, our LMCT model neglects the ligand hole in the LMCT ES that may influence the

effective charge at the Fe site and the ligand field splitting, thus slightly altering the LMCT ES spectral shape[19,21,35]. As a further limitation, our analysis of the Kβ main line region is unable to distinguish whether we observe the photoexcited $^2T_{2u}$ LMCT ES or the lower lying $^2T_{1u}$ LMCT ES that has been directly accessed in previous ultrafast studies using ~400 nm optical excitation[17,18,20], and could form rapidly in our experiment via internal conversion following photoexcitation.

To extract the time dependent populations of the LMCT ES and the second longer-lived transient species, denoted as *S2* in the following, we performed a fit of the normalized time dependent Kβ main line difference spectra $\Delta I_{K\beta}$ (Figs. 1c and 2a), using the following fit equation:

$$\Delta I_{K\beta}(t,E) = N_{LMCT}(t) \cdot (I_{K\beta}^{LMCT}(E) - I_{K\beta}^{ferric}(E)) + N_{S2}(t) \cdot \Delta I_{K\beta}^{S2}(E) \quad (2)$$

Here, $N_{LMCT}$ and $N_{S2}$ are the time dependent population fractions of the LMCT ES and the longer-lived transient component, respectively. $I_{K\beta}^{ferric}$ is the $^2[Fe^{III}(CN)_6]^{3-}$ ground state spectrum and $I_{K\beta}^{LMCT}$ is the $^1[Fe^{II}(CN)_6]^{4-}$ reference spectrum approximating the LMCT ES as described previously. We do not have reference spectra at hand that are suitable in terms of metal-ligand bonding to construct model difference spectra $\Delta I_{K\beta}^{S2}$ that would reliably distinguish between different spin and oxidation states of the longer-lived metal-cyanide transient species. Therefore, to extract the time dependent populations, we utilize the average difference spectrum in the 1.5 – 3 ps range to represent $\Delta I_{K\beta}^{S2}$. The VtC analysis (vide infra) independently confirms that the LMCT ES contribution to the difference spectrum can indeed be neglected after ~1.5 ps. However, very small differences between spectra averaged in the 1.5 – 3 ps and 20 – 40 ps time bins (Supplementary Note 1, Supplementary Fig. 1) suggest that multiple species or structural conformations may contribute to the extracted population fraction $N_{S2}$. Here, we neglect such subtle changes in favor of a more reliable fit procedure while keeping these limitations in mind. Further analysis of the transient VtC region (vide infra) is consistent with the presence of multiple spectroscopically similar species during the measured time window. Figures 2b and 2c show the extracted population fractions $N_{LMCT}$ and $N_{S2}$ together with kinetic fits using a multiexponential decay multiplied with a Heaviside step function and convoluted with a Gaussian instrument response function (Supplementary Note 3, Supplementary Tables 1 and 2). $N_{LMCT}$ grows within ~200 fs, the time resolution of our experiment, and decays completely with a fitted time constant of $253 \pm 27$ fs. The fit indicates a photoexcited LMCT ES fraction of $19 \pm 2$ %. Our LMCT ES time constant lies between the $^2T_{1u}$ LMCT ES lifetimes of ~0.5 ps, reported by Ojeda et al.[17] and the ~170 fs time constant reported by Engel et al.[18]. However, as pointed out earlier, from the Kβ main line analysis it remains unclear whether the $^2T_{2u}$ LMCT ES photoexcited in this work indeed relaxes into the $^2T_{1u}$ LMCT ES. Moreover, by comparing 265 nm and 400 nm LMCT ES excitation, Ojeda et al. reported a shortening of the LMCT ES lifetime at the higher excitation energy. Our ~250 fs time constant is consistent with this trend. Also, a mono-exponential decay is sufficient to fit the $N_{LMCT}$ population.

For $N_{S2}$, the absolute value cannot be associated with a population because we do not have an independently measured and properly normalized reference spectrum for the *S2* species. While the formation time of *S2* cannot be unambiguously resolved, the extracted population fraction peaks on a sub-picosecond timescale and then decays partially with an exponential time constant of ~3 ps (Fig. 2c). A small offset persists to 40 ps. Both, the ~3 ps partial decay timescale and the presence of a persistent offset point to the formation of a long-lived photoproduct. Specifically, photo-aquation as proposed by Fuller et al. should involve the production of transient $^2[Fe^{III}(CN)_5]^{2-}$ or $^4[Fe^{III}(CN)_5]^{2-}$ following cyanide anion abstraction[12]. However, conclusively assigning the Kβ main line difference spectrum associated with *S2* is challenging due to the lack of reliable reference spectra. After

~1.5 ps, the Kβ main line spectra are dominated by a blueshift of the $K\beta_{1,3}$ peak (Fig. 2a). While an increase in the effective spin moment could rationalize the observed blueshift of the $K\beta_{1,3}$ line, such a net shift could also reflect changes in metal-ligand equilibrium distances, covalency and ligand field strength, which significantly complicates the interpretation of these spectra[23,26,36]. Therefore, to assess the net effect of possible photoinduced chemical changes on the Kβ main line spectrum, we have performed quantum chemistry calculations (OpenMolcas, see Methods section) of geometry optimized structures for the $^2[Fe^{III}(CN)_6]^{3-}$, $^4[Fe^{III}(CN)_6]^{3-}$, square-pyramidal (SP) $^2[Fe^{III}(CN)_5]^{2-}$, SP $^4[Fe^{III}(CN)_5]^{2-}$ and $^2[Fe^{III}(CN)_5H_2O]^{2-}$ candidate species (Fig. 2d). To compare the calculated, broadened spectra in terms of their $K\beta_{1,3}$ peak positions, we aligned them to their center of mass which has experimentally been shown to vary by less than 0.1 eV for metal-cyanide compounds[27]. We then find that the $^4[Fe^{III}(CN)_6]^{3-}$, SP $^2[Fe^{III}(CN)_5]^{2-}$, SP $^4[Fe^{III}(CN)_5]^{2-}$ and $^2[Fe^{III}(CN)_5H_2O]^{2-}$ complexes all exhibit a blueshift of the $K\beta_{1,3}$ peak with respect to the $^2[Fe^{III}(CN)_6]^{3-}$ ground state, consistent with the experimentally observed blueshift. The calculated magnitude of the shift is largest for $^4[Fe^{III}(CN)_6]^{3-}$ (-0.55 eV) and amounts to 0.25 eV, 0.45 eV and 0.45 eV for the SP $^2[Fe^{III}(CN)_5]^{2-}$, SP $^4[Fe^{III}(CN)_5]^{2-}$ and $^2[Fe^{III}(CN)_5H_2O]^{2-}$ complexes, respectively. Not aligning the calculated spectra with respect to their center of mass changes the shift magnitudes but not their direction or order. While the calculated shift magnitudes need to be interpreted with caution, we note that experimentally for different iron-based compounds, a blueshift of ~0.5 eV of the $K\beta_{1,3}$ peak can be observed between ferric quartet and doublet species[37], in good agreement with our calculated shift between $^4[Fe^{III}(CN)_6]^{3-}$ and $^2[Fe^{III}(CN)_6]^{3-}$. The calculations also exhibit low intensities for the Kβ′ feature of these highly covalent metal-cyanide compounds, consistent with previous experimental findings on covalent compounds[38]. Consequently, the experimentally observed lack of significant intensity changes in the 7040–7050 eV range of the difference spectra after ~1.5 ps cannot be utilized to reliably reject the ferric quartet or sextet species as candidates for *S2*. This situation contrasts with less covalent ferric species that show an increase in Kβ′ intensities upon increase in spin multiplicity, while variations in the ligand environment resulted in significant $K\beta_{1,3}$ shifts with a lesser effect on Kβ′ intensities[26,27,37].

In summary, the Kβ main line analysis does not reliably determine the spin and oxidation state of the observed ~3 ps lived intermediate species. In the next section we show that the time dependent VtC difference spectra support a photo-aquation scenario proceeding via a ferric doublet penta-coordinate intermediate, while they likely exclude the occurrence of a transient quartet species.

**Femtosecond VtC x-ray emission spectra**

Figure 3a shows the time dependent VtC difference spectra averaged in different time bins (top inset) together with the experimental and calculated $^2[Fe^{III}(CN)_6]^{3-}$ ground state spectra (bottom inset). The $^2[Fe^{III}(CN)_6]^{3-}$ VtC spectrum gains intensity from Fe p-density admixed into ligand based molecular orbitals[39] and exhibits distinct groups of transitions[40–42]. The weak Kβ″ feature around 7095 eV arises from Lσ → Fe 1s transitions that exhibit strong sensitivity to the nature of the bound ligands. The transitions associated with the stronger $K\beta_{2,5}$ feature were assigned to predominantly Lσ → Fe 1s (~7106 eV) and Lπ → Fe 1s (~7109–7110 eV), respectively. Within the first ~300 fs, the observed difference spectrum strongly resembles the scaled difference of the $^1[Fe^{II}(CN)_6]^{4-}$ and $^2[Fe^{III}(CN)_6]^{3-}$ ground state spectra, identifying the LMCT ES fully consistent with the main line analysis. For both Kβ main line and VtC regions, the fitted scaling factors indicate similar LMCT ES population fractions of ~8% (Fig. 2a) and ~10% (Fig. 3a) within the 0.1–0.3 ps time bin, while the small residuals suggest the presence of *S2* at these early delays (Supplementary Note 2, Supplementary Fig. 2). The dominant feature of this transient VtC difference

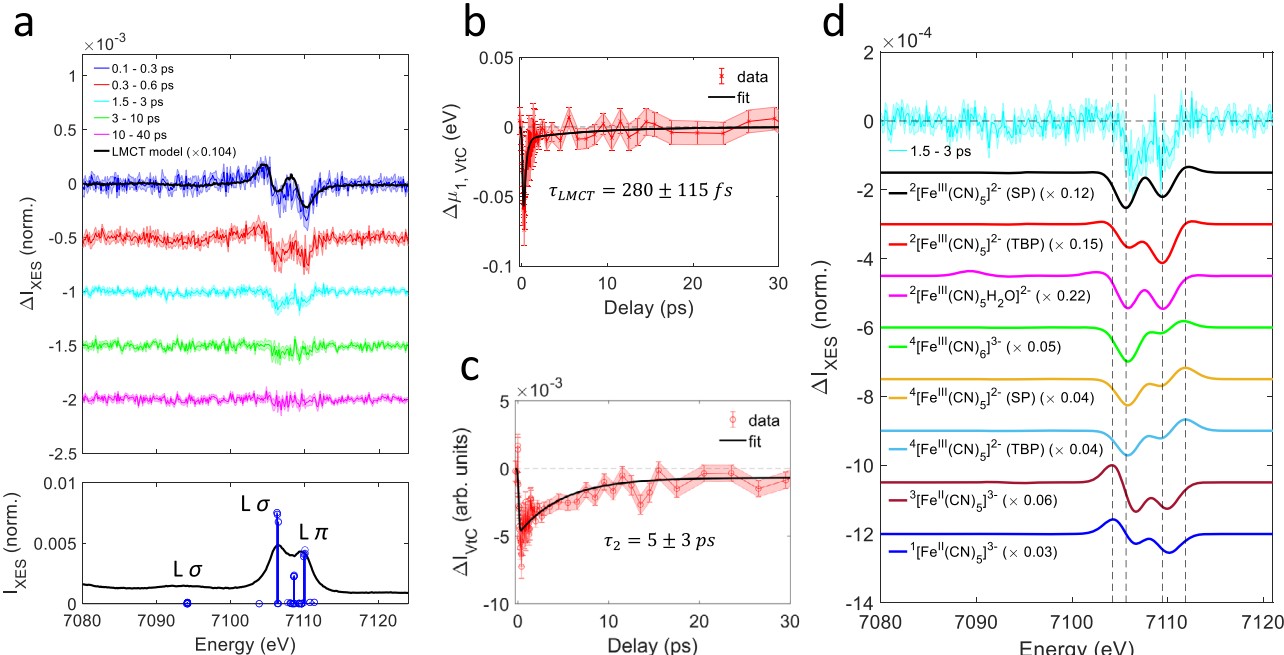

**Fig. 3 | Valence-to-core x-ray emission. a** The upper inset shows difference spectra in the valence-to-core (VtC) region for different time bins. Shaded areas reflect the standard deviation within a time bin when all difference spectra are rescaled to the bin-averaged summed difference signal magnitude. The black line (LMCT model) represents the difference between the ground state spectra of the aqueous $^1[Fe^{II}(CN)_6]^{4-}$ and $^2[Fe^{III}(CN)_6]^{3-}$ complexes. The lower inset shows the experimental $^2[Fe^{III}(CN)_6]^{3-}$ ground state spectrum (black) with the calculated transitions (blue). **b** Time dependence of the first moment position $\Delta\mu_{1,VtC}$ (red) with a kinetic fit (black). The estimation of uncertainties is described in Supplementary Note 4.

**c** Time dependence of the VtC intensity change $\Delta I_{VtC}$ (red) with a kinetic fit (black). The estimation of uncertainties is described in Supplementary Note 4. **d** Comparison between the 1.5−3 ps difference spectrum and calculated difference spectra for various candidate species as described in the text. Scaling factors of the calculated differences result from fitting the experimental difference in the 1.5−3 ps range and directly represent population fractions. Vertical dashed lines at 7104.23 eV, 7105.64 eV, 7109.43 eV, and 7111.84 eV are shown to facilitate comparisons between difference spectra.

spectrum is the overall shift towards lower emission energies due to a transient reduction of the Fe-site. After ~1 ps, the VtC difference spectrum is dominated by an overall intensity decrease which indicates an average reduction in metal-ligand orbital overlap[42], while no positive features persist in the difference spectra. We then calculate the time dependent VtC first moment shift $\Delta\mu_{1,VtC}(t)$ (Fig. 3b) and overall intensity change $\Delta I_{VtC}(t)$ (Fig. 3c) as described in Supplementary Note 4. The first moment shift tracks the metal spin and oxidation state while being less sensitive to the Fe-cyanide coordination[42]. Immediately after photoexcitation, we observe a decrease in $\Delta\mu_{1,VtC}$ due to the population of the LMCT ES. $\Delta\mu_{1,VtC}$ then recovers within a picosecond while a small negative signal persists beyond a picosecond. A bi-exponential kinetic fit (Supplementary Note 4, Supplementary Table 3) yields a fast decay constant of $280 \pm 115$ fs, in good agreement with the LMCT ES lifetime determined from the Kβ main line analysis. This decay accounts for more than 90% of the initial signal magnitude. The longer time constant cannot be reliably extracted from the fit.

For the VtC intensity change $\Delta I_{VtC}$, a kinetic fit (Supplementary Note 4, Supplementary Table 4) yields a partial exponential decay to a persistent offset within $5 \pm 3$ ps. Generally, the presence of multiple photochemical species and different sensitivities in the Kβ main line and VtC regions may give rise to different time evolutions of the difference signals. However, given the relatively large uncertainties of the kinetic fits, we are unable to robustly differentiate the ~5 ps timescale from the ~3 ps partial decay constant determined from the main line analysis.

To assign the prevailing species after the LMCT ES has decayed, we have performed ground state DFT calculations of the VtC spectra for various geometry optimized candidate species (Supplementary Note 4, Supplementary Figs. 4–6, Supplementary Tables 5 and 6) and then constructed simulated difference spectra with respect to the

$^2[Fe^{III}(CN)_6]^{3-}$ ground state. A comparison of the relevant calculated differences with the experimental difference spectrum in the 1.5−3 ps range is shown in Fig. 3d, a more detailed analysis including predicted first moment shifts is presented in Supplementary Note 4 (Supplementary Fig. 6). All calculated difference spectra reproduce the intensity decrease observed on the Kβ$_{2,5}$ peaks (7106−7109 eV) while additional features occur that we use to exclude some of these candidates. Reasonable fits are achieved for the penta-coordinate ferric doublet ($^2[Fe^{III}(CN)_5]^{2-}$) SP and trigonal bipyramidal (TBP) models, the ferric aquated complex ($^2[Fe^{III}(CN)_5H_2O]^{2-}$) and the penta-coordinate ferrous triplet model ($^3[Fe^{II}(CN)_5]^{3-}$). The ferrous singlet ($^1[Fe^{II}(CN)_5]^{3-}$) and the hexa- and penta-coordinate quartet models fit considerably worse. The quartet models exhibit an overall blueshift of the VtC transitions. However, the associated positive feature near 7112 eV and calculated increase in $\Delta\mu_{1,VtC}$ are not observed in the experiment. Both ferrous singlet and triplet penta-coordinate models exhibit a redshift that reflects the reduction of the Fe-site, but the negative contribution to $\Delta\mu_{1,VtC}$ predicted for these species exceeds the experimentally observed signal magnitude.

Based on these results, we assign the observed difference spectrum in the 1.5−3 ps range to a combination of the $^2[Fe^{III}(CN)_5]^{2-}$ SP and TBP intermediates and the $^2[Fe^{III}(CN)_5H_2O]^{2-}$ product. For SP $^2[Fe^{III}(CN)_5]^{2-}$, the calculations indicate a positive contribution to $\Delta\mu_{1,VtC}$, while for TBP $^2[Fe^{III}(CN)_5]^{2-}$ and $^2[Fe^{III}(CN)_5H_2O]^{2-}$, a slight negative contribution is predicted, in good agreement with the observed signal in the 1.5−3 ps range (Supplementary Fig. 6). The presence of these complexes is fully consistent with the Kβ main line analysis and the photo-aquation reaction proposed by Fuller et al.[12].

Due to the absence of significant spectral reshaping in the 1.5−40 ps range and the similarity of the calculated SP and TBP

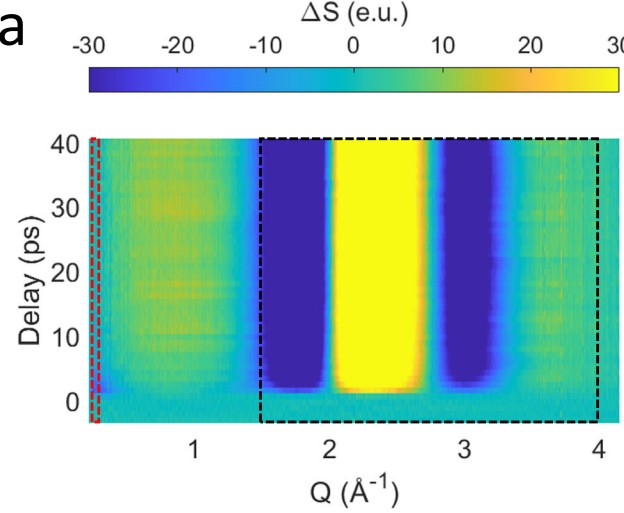

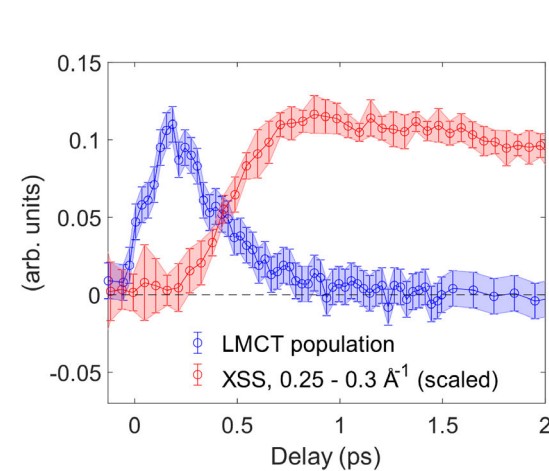

**Fig. 4 | X-ray solution scattering. a** Time dependent x-ray solution scattering (XSS) difference signal. The red box indicates the region used to extract the low-Q difference signal shown in **b**. The black box indicates the region used to extract the time dependent bulk water temperature increase. Source data are provided as a Source Data file. **b** Comparison of the ligand-to-metal charge transfer (LMCT) population fraction determined from the x-ray emission data (blue) and the scaled XSS low-Q difference signal (red) collected with small time bins to resolve the signal rise time. For the LMCT population fraction, the uncertainty at each time delay was estimated using a cutoff in the increase of the sum of squared residuals when varying the population fraction with respect to the optimized value. For the XSS low-Q difference signal, uncertainties reflect the standard deviation of kinetic traces within the 0.25–0.3 Å$^{-1}$ range, when all traces are rescaled to the mean summed difference signal magnitude within that Q-range.

$^2$[Fe$^{III}$(CN)$_5$]$^{2-}$ and $^2$[Fe$^{III}$(CN)$_5$H$_2$O]$^{2-}$ difference spectra in the VtC region (Fig. 3d), we are unable to quantify the relative contributions and interconversion pathways of these species. In fact, the observed 1–5 ps partial decay observed in the combined Kβ main line and VtC region supports a scenario in which the penta-coordinate intermediate decays either through recombination with cyanide anions or uptake of a water molecule to form the $^2$[Fe$^{III}$(CN)$_5$H$_2$O]$^{2-}$ product, possibly alongside some geometric conversion between the SP and TBP structures. An analogous scenario applies to the photo-aquation mechanism of $^1$[Fe$^{II}$(CN)$_6$]$^{4-}$ [34,43], however involving intersystem crossing to form $^3$[Fe$^{II}$(CN)$_5$]$^{3-}$, rather than $^1$[Fe$^{II}$(CN)$_5$]$^{3-}$. Here for the ferric case, our DFT calculations predict $^4$[Fe$^{III}$(CN)$_5$]$^{2-}$ as the lowest energy penta-coordinate complex but the VtC analysis clearly favors the $^2$[Fe$^{III}$(CN)$_5$]$^{2-}$ complexes instead. We note however that the $^2$[Fe$^{III}$(CN)$_5$]$^{2-}$ complex is only slightly higher in energy than $^4$[Fe$^{III}$(CN)$_5$]$^{2-}$ and should remain energetically well accessible prior to significant solute excess energy dissipation (Supplementary Table 5).

## X-ray solution scattering difference curves

Figure 4a shows the time dependent XSS difference signal that has been recorded simultaneously with the Kβ main line and VtC XES differences using the setup shown in Fig. 1b. This signal reflects the photoinduced structural response of both solute and solvent constituents. Within the first few picoseconds, the difference signal in the low-Q range (<0.5 Å$^{-1}$) exhibits a negative feature reflecting a transient decrease in scattering intensity. A comparison with the time dependent LMCT ES population from the XES analysis (Fig. 4b) indicates that the onset of the low-Q XSS difference signal is delayed and only occurs as the LMCT ES decays. The absence of a significant structural response directly associated with the LMCT ES is consistent with the modest intramolecular structural rearrangement previously determined for the $^2$T$_{1u}$ LMCT ES[12]. Furthermore, this seems to indicate that there is negligible structural reorganization of the solute-solvent atom pair distances in the LMCT ES with respect to the ground state, but assessing the impact of the charge redistribution on the solvation shell lies outside the scope of this work. A mono-exponential fit indicates that the low-Q XSS difference signal decays with a time constant of 4.4 ± 1.7 ps (Supplementary Note 5, Supplementary Fig. 7), similar to

the timescales determined from the Kβ main line and VtC XES analysis. The observed negative low-Q XSS difference signal is indicative of a reduction in the total electron density[30,44], and therefore consistent with the loss of a cyanide ligand and average elongated Fe-cyanide bond distances of the proposed $^2$[Fe$^{III}$(CN)$_5$]$^{2-}$ complexes. Importantly, the delayed onset of the low-Q XSS difference signal suggests that Fe-cyanide bond expansion is primarily a consequence of the LMCT ES decay rather than associated with the direct photoexcitation of the MC ES.

We now focus on the 1.5–4.0 Å$^{-1}$ range of the time dependent XSS difference signal (Fig. 4a, Supplementary Note 5, Supplementary Fig. 7,), which is dominated by signatures arising from bulk water heating and density changes. As detailed in Supplementary Note 5, we utilize this difference signal to extract the time dependent increase in bulk water temperature. The water temperature stabilizes after 10–20 ps reaching a maximum increase of $\Delta T \approx 1.66$ K. This indicates that most of the solute vibrational excess energy has dissipated at this point. Given the 336 nm pump wavelength and known sample concentration, we then estimate the total photoexcitation fraction as $f_{exc} \approx 20\%$ (Supplementary Note 5). This agrees with the photoexcited 19 ± 2% LMCT ES population fraction independently determined from the XES measurement, implying a small direct photoexcitation fraction for the symmetry-forbidden $^2$A$_{2g}$ and $^2$T$_{2g}$ MC ES located in the 309–336 nm range[15,16]. In contrast, fitting the simulated VtC XES differences of the $^2$[Fe$^{III}$(CN)$_5$]$^{2-}$ complexes to the experimental difference spectrum averaged in the 1.5–3 ps window requires significantly higher population fractions of ~12–15% (Fig. 3d). This suggests a photolysis quantum yield close to unity, given the ~20% photoexcitation fraction and 1–5 ps decay of the photo-aquation reaction intermediates. Moreover, the temporal onset of the XSS low-Q difference signal indicates that structural changes associated with the ligand dissociation process occur following the decay of the LMCT ES. We therefore conclude that the observed penta-coordinate population fraction predominantly derives from LMCT rather than MC ES photoexcitation. As proposed by Ojeda et al.[17], direct photolysis from the $^2$T$_{1u}$ LMCT ES may produce the ferrous photo-aquation

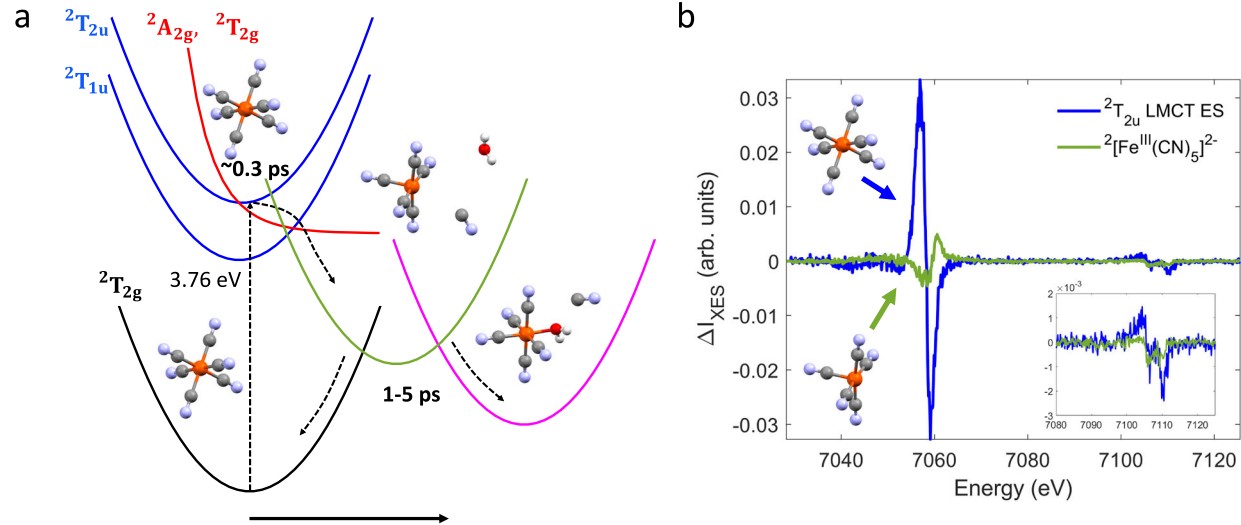

**Fig. 5 | Proposed scenario for the photo-aquation reaction of aqueous ferricyanide. a** Proposed relaxation scheme for 336 nm photoexcitation of aqueous $^2[Fe^{III}(CN)_6]^{3-}$. Photoexcitation of the $^2T_{2g}$ electronic ground state predominantly populates the $^2T_{2u}$ Ligand-to-metal charge transfer (LMCT) excited state (ES). Metal-to-ligand back electron transfer occurs within ~0.3 ps and involves population of the $^2A_{2g}/^2T_{2g}$ metal-centered ES which facilitate cyanide dissociation and

formation of the penta-coordinate $^2[Fe^{III}(CN)_5]^{2-}$ intermediate. Uptake of a water molecule competes with geminate recombination within a few picoseconds. **b** Species associated difference spectra for the $^2T_{2u}$ LMCT ES (blue) and $^2[Fe^{III}(CN)_5]^{2-}$ intermediate (green) extracted from a global target analysis of the combined Fe Kβ main line and valence-to-core range.

products alongside cyanide radicals. However, we did not detect those photoproducts in our measurement and therefore propose an alternative explanation that invokes rapid nonradiative deactivation of the photoexcited LMCT ES through the energetically close lying MC ES that facilitates cyanide dissociation (Fig. 5a). While we do not have direct evidence for the population of these states, the small energy gap between the $^2T_{2u}$ LMCT and MC ES and appreciable charge transfer character in the MC ES[15] may enable efficient coupling that prevents relaxation into the lower lying $^2T_{1u}$ LMCT ES. A similar mechanism has been proposed to initiate CO-photolysis from gas-phase $Cr(CO)_6$[45]. The presence of MC ES often rationalizes the observed short LMCT ES lifetimes of ferric molecular complexes[8,46]. Here, the MC ES have an electron in the Fe-CN antibonding $e_g$-orbital, thus likely triggering Fe-cyanide bond dissociation similar as observed in the ferrous case[34]. However, the difference in reported photo-aquation quantum yields for $^1[Fe^{II}(CN)_6]^{4-}$ (10–20%)[47] and $^2[Fe^{III}(CN)_6]^{3-}$ (2–6%)[12] requires some clarifications. For 336 nm excitation of aqueous $^2[Fe^{III}(CN)_6]^{3-}$, we propose that both direct and indirect population of the MC ES via the $^2T_{2u}$ LMCT ES can contribute to ligand photolysis. Therefore, the photo-aquation quantum yield is not reduced by the branching between the LMCT and MC ES in the photo-excitation process. Instead, we hypothesize that the overall lower quantum yields reported in the ferric case arise from larger recombination fractions of $^2[Fe^{III}(CN)_5]^{2-}$ with cyanide anions, consistent with lower electrostatic repulsion between these species.

The proposed absence of a transient spin state change during the $^2[Fe^{III}(CN)_6]^{3-}$ photo-aquation reaction is another remarkable difference with respect to the ferrous case, where aquation was observed on a ~20 ps timescale[34]. The lack of a spin barrier could enable rapid solvent coordination as observed for iron carbonyl photolysis in ethanol[48], however the limited sensitivity of our XES and XSS data does not allow to determine the formation timescales of $^2[Fe^{III}(CN)_5H_2O]^{2-}$. Furthermore, the effect of transient spin state changes in the presence of large amounts of solute vibrational excess energy that could stabilize the penta-coordinate intermediate, remains unclear[34,49].

## Global target analysis of the combined Fe Kβ main line and VtC x-ray emission difference spectra

Using a singular value decomposition (Supplementary Note 4, Supplementary Fig. 3) and the proposed kinetic scheme for the photo-aquation reaction (Fig. 5a), we have also extracted species associated difference spectra (SADS) in the combined Fe Kβ main line and VtC range. Details of the analysis can be found in Supplementary Note 6 (Supplementary Fig. 8). The extracted SADS for the $^2T_{2u}$ LMCT ES and the penta-coordinate intermediate are shown in Fig. 5b. The SADS for the $^2T_{2u}$ LMCT ES resembles the measured difference spectrum in the 0.1–0.3 ps range (Figs. 2a and 3a), consistent with the predominant presence of this species in that time bin. The SADS of the penta-coordinate intermediate clearly shows a blueshift in the main line region and an overall intensity reduction in the VtC region, as indicated by the XES difference spectra after ~1.5 ps (Figs. 2a and 3a). Both spectral features have been rationalized by our calculations for the $^2[Fe^{III}(CN)_5]^{2-}$ intermediate (Figs. 2d and 3d). Moreover, comparing the SADS with the DFT-based calculated differences, in both shape and magnitude, clearly favors $^2[Fe^{III}(CN)_5]^{2-}$ over $^4[Fe^{III}(CN)_5]^{2-}$ (Supplementary Fig. 8), which further supports the proposed scenario.

## Discussion

The ES dynamics of aqueous ferricyanide has been a controversial topic with conflicting reports on intramolecular relaxation and primary photochemical reaction products. Using 336 nm excitation predominantly into the $^2T_{2u}$ LMCT ES combined with femtosecond hard x-ray probes, we unambiguously resolve the sub-ps lifetime of the $^2T_{2u}$ LMCT ES and propose a deactivation pathway via close lying dissociative MC ES, thus we rationalize its ultrashort lifetime. We also report previously undetected transient intermediates that are formed following decay of the $^2T_{2u}$ LMCT ES. By leveraging our novel experimental methodology that combines Fe Kβ main line and VtC XES analysis, we assign these species to the $^2[Fe^{III}(CN)_5]^{2-}$ photo-aquation reaction intermediate and the $^2[Fe^{III}(CN)_5H_2O]^{2-}$ product. The low reported photo-aquation quantum yields for aqueous $^2[Fe^{III}(CN)_6]^{3-}$ are assigned to efficient geminate recombination between the penta-coordinate intermediate and the cyanide anions. The proposed scenario contrasts with previous studies of aqueous $^2[Fe^{III}(CN)_6]^{3-}$ utilizing

400 nm excitation[17,18], since we do not find evidence of a short-lived quartet MC ES or the $^1[Fe^{II}(CN)_5H_2O]^{3-}$ complex. This difference may arise from the higher excitation wavelength used in our study that enables relaxation into the $^2A_{2g}$ and $^2T_{2g}$ MC ES thus opening an efficient channel for cyanide dissociation that is inaccessible using 400 nm excitation. The increase in photoproduct fraction found by Ojeda et al[17]. when using 265 nm rather than 400 nm excitation and recent picosecond x-ray studies[33] support a wavelength dependent dissociation mechanism.

These findings contrast with the photo-aquation mechanism of $^1[Fe^{II}(CN)_6]^{4-}$: Here, the dominant contribution to cyanide photolysis stems from symmetry-allowed excitation of an LMCT ES that efficiently relaxes into dissociative MC ES rather than from direct excitation of weak symmetry-forbidden MC ES. Furthermore, cyanide photolysis is not accompanied by intersystem crossing and the entire photo-aquation reaction occurs via a ferric doublet penta-coordinate intermediate. The absence of a spin state change circumvents significant structural relaxation of the penta-coordinate intermediates, but it remains unclear how the absence of a spin barrier impacts the reaction kinetics, which may be influenced by excess energy dissipation from the solute into the solvent.

Finally, the dynamics of bond photolysis and reformation and their role in photocatalytic processes has been studied in a wide range of solvated transition metal complexes in solution including noble or abundant metal-based, biological, and bioinspired complexes. Understanding reaction outcomes based on the multitude of influencing factors requires methods sensitive to electronic, spin and structural observables. While femtosecond Fe Kβ main line spectroscopy remains a powerful probe to track spin and oxidation state changes, we demonstrate that the combination with VtC XES enables a more reliable identification of elusive transient reaction intermediates and photoproducts via enhanced sensitivity to the metal-ligand nuclear structure and bonding. This work focuses on open questions in the photo-physics and chemistry of the aqueous $^2[Fe^{III}(CN)_6]^{3-}$ model compound, but the extension of this methodology towards solvated transition metal active sites embedded in more complex environments of photo-catalytically relevant systems will be particularly powerful due to the site-selective nature of the x-ray probe. Emerging capabilities at x-ray free electron lasers could for instance enable deeper studies of photoinduced active site dynamics in dilute, biologically relevant systems such as heme proteins.

## Methods
### Experimental setup
The Kβ main line and VtC XES data were collected at the X-ray Correlation Spectroscopy (XCS) instrument at the Linac Coherent Light Source (LCLS)[50]. The sample was flowed through a 50 µm diameter cylindrical liquid jet, using an HPLC pump. We used a 100 mM aqueous potassium hexacyanoferrate(III) ($K_3Fe(CN)_6$, purchased from Sigma-Aldrich) solution to obtain an absorbance of ~0.37 at the excitation wavelength. The sample was optically pumped and probed by 8.5 keV self-amplified stimulated emission (SASE) x-ray pulses (~$5 \cdot 10^{11}$ photons/pulse at the sample, 120 Hz, ~40 fs, $\Delta E/E$ ~$5 \times 10^{-2}$) shortly after exiting the capillary in the region of laminar flow. Optical excitation was performed nearly collinearly to the x-rays with ~50 fs FWHM, 336 nm laser pulses (~3.6 µJ) with ~100 µm diameter generated from an optical parametric amplifier pumped by the output of a Ti:sapphire regenerative amplifier laser system (Coherent, Legend). The pump laser fluence was chosen to maximize the excited-state fraction while avoiding multiphoton absorption effects. The time delay between the laser and x-ray pulse was determined via the timing tool[51] installed at XCS. The x-ray pulses were focused using Be compound refractive lenses to a ~20 µm diameter spot size on the sample jet. A high-energy resolution x-ray emission spectrometer, based on the von Hamos geometry, was used to capture the Fe Kβ main line and VtC XES

signal[52]. The spectrometer was equipped with 4 cylindrically bent (0.25 m bending radius) $110 \times 25$ mm$^2$ Ge(620) crystal analyzers and set to cover the Bragg angle range from 76.6° to 80.4° corresponding to an energy range of 7.030 to 7.125 keV. The energy resolution is estimated to be ~0.6 eV[42]. The XES data were collected using an ePix100 detector[53]. A helium bag was used between the sample, crystals and detector to minimize attenuation of the fluorescence from air and reduce background from diffusely scattered radiation. Full 2D images of the XES detector were read out shot-to-shot and subsequently processed and binned according to their pump-probe delay. XES spectra were extracted by integrating the intensity in two rectangular areas of interest each containing a few pixels along the non-dispersive axis. The emission energy was calibrated by comparing the laser off spectra to a previously measured reference spectrum of the same compound. All measured spectra were normalized to the total area of the Kβ main line x-ray emission signal.

### Calculation of the Kβ main line x-ray emission spectra
Fe Kβ main line x-ray emission spectra were calculated using the restricted active space (RAS) method with OpenMolcas[54]. The metal 3p core orbitals are placed in subspace RAS1 and the 1s core orbital is placed in RAS3. The five metal 3d character orbitals together with two ligand character s donation orbitals and three empty p orbitals are placed in the RAS2. The 1s and 3p core ionized states are calculated with a novel projection technique called HEXS[55,56] that sets the configuration interaction coefficients of configuration state functions with doubly occupied core-orbitals to zero. The dynamic correction is treated at the level of second-order perturbation theory (CASPT2) using the multi-state formalism[57]. Scalar relativistic effects have been included by using a second-order Douglas-Kroll-Hess Hamiltonian[58,59], in combination with the ANO-RCC-VDZP basis set and the use of a Cholesky decomposition approach to approximate the two-electron integrals[60,61]. The electric dipole oscillator strengths including the spin-orbit coupling is calculated with the RAS state interaction approach. The calculated x-ray emission spectra are aligned to the same center-of-gravity position, then shifted by another 1.1 eV to align the peaks of the experimental and calculated $^2[Fe^{III}(CN)_6]^{3-}$ spectra[62].

### Calculation of the valence-to-core x-ray emission spectra
Density functional theory calculations were performed using the ORCA 4.2.1 package[63]. Geometry optimization for different species was carried out using the B3LYP functional, def2-TZVP basis set[64] and the DFT-D3 dispersion correction with Becke-Johnson damping[65]. The effect of the solvent was considered by using the conductor-like polarizable continuum model (C-PCM)[66] for water. The calculated $^2[Fe^{III}(CN)_6]^{3-}$ and $^1[Fe^{II}(CN)_6]^{4-}$ structures are in close agreement with experimentally reported structures[67]. The VtC x-ray emission spectra were calculated using the one-electron approach described by Lee et al.[37]. Only dipole transitions were included in the spectrum. The B3LYP functional and ZORA-def2-TZVP basis set were used, except for the Fe atom, where the core properties basis set CP(PPP) has been used with a special integration accuracy of 7[37,41,42,68]. Scalar relativistic effects were considered via the zero-order regular approximation (ZORA)[69]. The calculated transitions were broadened by a 3.0 eV FWHM Gaussian function and shifted by 23.035 eV to overlap with the experimental spectra.

## Data availability
The XES and XSS data shown in Figs. 1c and 4a are provided as Source Data files. Source data are provided with this paper.

## Code availability
All relevant data and analysis scripts used in this study are available from the corresponding authors upon request.

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

## Acknowledgements

This work was supported by the U.S. Department of Energy, Office of Science, Basic Energy Sciences, Chemical Sciences, Geosciences, and Biosciences Division. Use of the Linac Coherent Light Source (LCLS), SLAC National Accelerator Laboratory, was supported by the U.S. Department of Energy, Office of Science, Office of Basic Energy Sci-ences under Contract No. DE-AC02-76SF00515. Use of the Stanford Synchrotron Radiation Lightsource, SLAC National Accelerator Labora-tory, is supported by the U.S. Department of Energy, Office of Science, Office of Basic Energy Sciences under Contract No. DE-AC02-76SF00515. DFT calculations were supported with resources from the SSRL Structural Molecular Biology Program supported by the DOE Office of Biological and Environmental Research, and by the National Institutes of Health, National Institute of General Medical Sciences (P41GM103393). The contents of this publication are solely the respon-sibility of the authors and do not necessarily represent the official views of NIGMS or NIH. This research used resources of the National Energy Research Scientific Computing Center (NERSC), a U.S. Department of Energy Office of Science User Facility operated under Contract No. DE-AC02-05CH11231. E.B acknowledges support by the US Department of Energy, Office of Science, Basic Energy Sciences, Chemical Sciences, Geosciences, and Biosciences Division, Condensed Phase and Inter-facial Molecular Science program, FWP 16248. A.G. was supported by U.S. Department of Energy, Office of Science, Office of Basic Energy Sciences, Chemical Sciences, Geosciences, and Biosciences Division, Catalysis Science Program to the Ultrafast Catalysis FWP 100435.

## Author contributions

D.S., R.A.M., A.G. T.C.W. and M.R. designed the research and experi-ments. M.R., A.G., A.G.E., E.B., M.Q., A.B., K.L., K.K., C.W., T.v.d., J.R., J.M.G., T.K. conducted the experiment at the LCLS. M.R., T.K., A.G., E.B., K.G. and D.S. analyzed and interpreted the data. M.R. performed DFT calculations with help from K.L. M.G. performed OpenMolcas calcula-tions. M.R., D.S., and R.A.M. wrote the manuscript with input from all authors.

## Competing interests

The authors declare no competing interests.
