## [Peer Review File · Nature Communications]

Ferricyanide photo-aquation pathway revealed by combined femtosecond $K\beta$ main line and valence-to-core x-ray emission spectroscopyReviewer #1 (Remarks to the Author):

The manuscript describes the application of pump-probe X-ray emission and solution scattering techniques to a rather simple model complex $[\text{Fe}(\text{CN})_6]^{3-}$. The performed experiments fully benefit from advanced properties of LCLS free electron laser such as high flux and temporal resolution, which I see as the strength of the manuscript. In particular, pump-probe valence-to-core X-ray emission spectra were measured, which was a challenge in the pump-probe X-ray community for years. Nevertheless, such measurements for very similar complex $[\text{Fe}(\text{CN})_6]^{4-}$ were reported already by the same group (J. Chem. Phys. 152, 074203 (2020)). The combination of X-ray emission spectroscopy with solution scattering was also reported for the related system containing $\text{Fe}(\text{CN})_6$ structural unit (Nature Chemistry 13, 343). Therefore from the technical point of view, there is no breakthrough in the current work. Analyzing the photochemical results, I'm convinced by the reported observations and conclusions about the ligand-to-metal charge transfer (LMCT) state. A significant part of the manuscript is about short-lived species formed after the LMCT state. Regarding this part, I'm not so convinced and this part looks for me at the edge of observations and speculations. In particular, the shape of the transient valence-to-core signal for 1.5-3 ps time bin is so similar to the inverted valence-to-core spectrum that I cannot fully exclude that it appears as the artifact of the data analysis. To get rid of these doubts I suggest to fully benefit from the fact that valence-to-core and main line XES spectra are measured simultaneously rather than analyzing them independently as presented in the current version of the manuscript. In supplementary note 4, the authors show the results of singular value decomposition (SVD) of their time-resolved XES data. First of all, for me it was a strong argument supporting that additional states beyond LMCT are actually observed. SVD-based results can be shown in the main text (in the current version of the main text SVD is not mentioned at all.) Most importantly, by the transformation of SVD components (using a linear combination of them with coefficients which are exactly the same for the main line and the valence-to-core region) one should be able to find a model which is in agreement with both expectations from theory for the main line and for the valence-to-core region.

Regarding the solution scattering part of the manuscript, it is unclear why only the response of the solvent is observed in the data. Authors already have structural models from DFT for different states and some estimates of their fraction from XES analysis. On this basis, they can calculate the expected contribution from the solute to the shape of the pump-probe scattering signal and compare it with the experimental data. Is it really so weak signal that it goes below the noise?

The excitation energy used during the experiment is significantly above the energy of the absorption edge. At these conditions, KL-beta satellites appear which are very close in energy to the valence-to-core peak and have comparable intensity. These peaks should be also considered for the data analysis.

The intensity of the K-beta'' peak in the theoretical calculations for the ground state is significantly lower than in the experiment. It is unclear why and it adds some doubts regarding the method (or selected parameters) used to calculate the spectra. Is it possible to achieve better agreement of the intensities? How far are the relative intensities of Kbeta_{1,3} and Kbeta_{2,5} peaks in the theory and the experiment?

In the article Nature Chemistry 13, 343 (which has significant overlap in the list of co-authors with this manuscript) for the system with a similar structure around the Fe center the importance of solute-solvent interactions was demonstrated. Why this effect is negligible for $[\text{Fe}(\text{CN})_6]^{3-}$?

Summarizing my overall impression from the manuscript: the experimental data I see as another good example of pump-probe measurements with an advanced LCLS instrument. Regarding the data interpretation, the authors are on a good track, but the analysis can be improved to make it convincing.

Reviewer #2 (Remarks to the Author):

Overview:

This is the review of "Ferricyanide photo-aquation pathway revealed by combined femtosecond K β main line and valence-to-core x-ray emission spectroscopy" by Reinhardt et al. This work describes novel ultrafast X-ray emission measurements of the photodissociation/aquation of ferricyanide. The work is technically impressive and the conclusions are clear and validated. I recommend publication in Nature Communications with the following considerations.

Accomplishments:

1. VtC XES in combination with main-edge XES with sufficient signal to noise to draw conclusions is worthy of publication in Nature Communications. Correlation of the changes in VtC XES with those expected for the main-edge XES is an excellent means of validating both the experiment and the interpretation of the result, as these measurements are complementary.
2. The use of singular value decomposition (SVD) of the time resolved signal is an excellent approach to extract the dominant spectral changes and associated chemical species observed and mimics analysis in previous works, such as Ref 63.
3. Additional combination with solvent scattering data is also interesting, but outside my area of expertise.
4. The combination with restricted active space and DFT calculations of the two XES spectra is excellent and their integration into the analysis is appropriate.

Criticisms:

1. There may be significant differences in the estimated relative energies of the various complexes when the first solvation shell of water molecules is included due to the presence of hydrogen-bonding between these molecules and the CN anions. Lack of H-bonds in the SP and TBP intermediates might raise their energies significantly. However, it is likely that the spectral simulations are not dramatically impacted by lack of inclusion of the solvating waters. The authors should comment on how much their analysis might differ if these details of solvation were considered.

Specific details to aid readability:

1. Please provide an explanation of the spin-multiplicity notation for the various cyanide complexes - indicating that the first superscript number refers to the spin multiplicity (doublet, quartet, etc.) - not all previous publication have used this compact notation before and explaining it the first time it appears will benefit the general reader.

Reviewer #3 (Remarks to the Author):

The manuscript by Reinhardt et al. presents a thorough analysis of the relaxation pathway of photoexcited aqueous ferricyanide anions using ultrafast x-ray emission spectroscopy. These metallic complexes exhibit complicated relaxation pathways upon photoexcitation and most techniques offer an incomplete window to characterize them. Reinhardt et al. uses the exciting capabilities of LCLS to implement fs XES that captures the main K β transition, but also extend to higher energies to measure also the valence to core transitions. They also complete their approach using an XSS signal. This allows them to propose a sound relaxation pathway (excitation in LMCT states, then population of MC states and finally dissociation and aquation) while expliciting clearly the limits of their technique (evidence of the transfer to MC states).

The overall analysis is nicely written and supports well the claims. The discussion and

comments on previous results in the literature (Ojeda et al, Engel et al) also add a good comparison with the established literature. The authors do a good work at showing that the Kb shift cannot be used to unambiguously determine the oxidation and spin of the intermediate species, complemented by quantum chemistry arguments.

Small remarks and questions that came to me while reading the manuscript are:

- the authors use a combination of post-Hartree Fock theory (RAS SCF + CASPT2) for the Kb and density functional theory for the valence to core emission. Why not using RASSCF for both? I assume that the size of the active space becomes impractical, but could the limitations of DFT + one electron approach for VtC transitions bring some issues? Does the different level of theory used for different part of the spectra complicates the analysis and merging of spectra?
- It would be convenient to write explicitly in SI the fitting equations used for different models (not necessarily all of them) to compare with the summarizing tables at a glance. Now they can be deduced from the text.
- Some comments could be added on how error bars are computed, e.g. in Fig 2a and b (standard deviation over some number of shots, propagated uncertainty, enlarged or not to account for a confidence interval, etc). Similarly, uncertainties of the fitted parameters could be described more (are the datapoints weighted by their experimental uncertainty in the fitting routine, how are calculated the uncertainties of the parameters of the nonlinear fitting model, etc).
- the article concludes with an opening toward general transition metal sites in complex environments. The authors could comment on the applicability of the technique to study proteins, for example small molecule dissociation in heme proteins or other systems that interest them.

I find the work to show important advances and recommend its publication in Nature Communications. The ferricyanide complex is an interesting case to study on its own, and the methodology can be applied to a vast range of metallic complexes.

Point-by-point response to reviewer comments

Reviewer 1:

A significant part of the manuscript is about short-lived species formed after the LMCT state. Regarding this part, I'm not so convinced and this part looks for me at the edge of observations and speculations. In particular, the shape of the transient valence-to-core signal for 1.5-3 ps time bin is so similar to the inverted valence-to-core spectrum that I cannot fully exclude that it appears as the artifact of the data analysis. To get rid of these doubts I suggest to fully benefit from the fact that valence-to-core and main line XES spectra are measured simultaneously rather than analyzing them independently as presented in the current version of the manuscript. In supplementary note 4, the authors show the results of singular value decomposition (SVD) of their time-resolved XES data. First of all, for me it was a strong argument supporting that additional states beyond LMCT are actually observed. SVD-based results can be shown in the main text (in the current version of the main text SVD is not mentioned at all.) Most importantly, by the transformation of SVD components (using a linear combination of them with coefficients which are exactly the same for the main line and the valence-to-core region) one should be able to find a model which is in agreement with both expectations from theory for the main line and for the valence-to-core region.

We thank the reviewer for critically reviewing our XES data analysis. While we believe that the data shows robust evidence of additional species beyond the identified short-lived LMCT ES, we agree that a combined analysis of the main line and VtC regions could make this point more convincing and the spectral assignments more apparent: The 1.5 – 3 ps difference spectrum in the valence-to-core region indeed resembles the inverted ground state spectrum, while the main line region shows a blueshift instead. Since data reduction has been performed in an identical fashion for the entire XES spectral range, we believe that a significant contribution of data analysis artifacts to the interpreted difference signal can be safely excluded. Moreover, these spectral features for both the main line and VtC regions have been rationalized using theory and the observed decrease in intensity in the VtC region is expected when dissociating a cyanide ligand.

To further strengthen these points, we have performed a global target analysis as suggested by the reviewer. An additional section in the manuscript and Supplementary Note 6 have been added to provide details of this analysis. Specifically, we have extracted species associated difference spectra (SADS) for the LMCT ES and the species proposed to be involved in the photo-aquation reaction in the combined Fe K β main line and VtC spectral range. We have first reconstructed / noise-filtered the XES difference map using two components, then performed a global fit that yields time constants consistent with our separate analysis of the K β main line and VtC spectral ranges. These time constants were then used with a kinetic scheme of the photo-aquation reaction to extract the SADS. The SADS of the penta-coordinate intermediate

confirms the previously presented analysis, showing a blueshift of the Fe K β main line and overall intensity decrease in the VtC region with respect to the $^2[\text{Fe}^{\text{III}}(\text{CN})_6]^{3-}$ ground state. In addition, the magnitude of the SADS of the penta-coordinate complex in the VtC region is most consistent with the DFT-based, calculated difference spectra of the doublet penta-coordinate intermediates ($^2[\text{Fe}^{\text{III}}(\text{CN})_5]^{2-}$) while the quartet penta-coordinate ($^4[\text{Fe}^{\text{III}}(\text{CN})_5]^{2-}$) complexes show significantly larger magnitudes.

While updating the manuscript, we have largely kept the discussion of the individual XES ranges and the XSS difference signal and inserted the combined analysis afterwards. We hope that this choice of presenting the results emphasizes the construction of the kinetic model used for the global target analysis.

Regarding the solution scattering part of the manuscript, it is unclear why only the response of solvent is observed in the data. Authors already have structural models from DFT for different states and some estimates of their fraction from XES analysis. On this basis, they can calculate the expected contribution from the solute to the shape of the pump-probe scattering signal and compare it with the experimental data. Is it really so weak signal that it goes below the noise?

The XSS section of the manuscript primarily focuses on the extraction of the excitation fraction from the bulk water heating signal. In addition, a small low-Q negative difference signal is distinguished from the bulk water heating signal and assigned to structural changes associated with the photo-aquation reaction due to its delayed appearance and longer lifetime with respect to the LMCT ES population. However, reliably extracting structural information from this limited Q-range is challenging.

The difference scattering signal can be expressed as $\Delta S(Q, t) = \Delta S_{\text{solute-solute}}(Q, t) + \Delta S_{\text{solute-solvent}}(Q, t) + \Delta S_{\text{solvent-solvent}}(Q, t)$ (Biasin *et al.*, Nature Chemistry 13, 343-349 (2021)). As requested by the reviewer, we have estimated the solute-only contribution $\Delta S_{\text{solute-solute}}$ of the proposed species to the total XSS difference signal and added it as Supplementary Note 7. These calculations indeed suggest small, but non-negligible solute-only differences. As may be expected for a small solute that contains only a few atoms, strongly interacts with the solvent and undergoes a chemical reaction explicitly involving solvent molecules, the calculated $\Delta S_{\text{solute-solute}}$ does not reproduce the observed low-Q difference signal, which is significantly influenced by $\Delta S_{\text{solute-solvent}}$. Reliably calculating the $\Delta S_{\text{solute-solvent}}$ term would require extensive modeling efforts including high level molecular dynamics simulations and this lies outside the focus of this work.

The excitation energy used during the experiment is significantly above the energy of the absorption edge. At these conditions, KL-beta satellites appear which are very close in energy to the valence-to-core peak and have comparable intensity. These peaks should be also considered for the data analysis.

We agree with the reviewer that for the incident x-ray energy used in our measurements (8.5 keV) the creation of multiple core holes may result in satellite contributions to the measured spectra. To assess the potential impact on our measurements we have compared our $^2[\text{Fe}^{\text{III}}(\text{CN})_6]^{3-}$ ground state spectrum with that measured by Ross *et al.* (J. Phys. Chem. B 2018, 122, 19, 5075-5086) for the same compound. Importantly, Ross *et al.* have used a significantly lower incident x-ray energy of 7.5 keV, insufficient to create K β L satellites. The similarity of our spectrum with that reported by Ross *et al.* indicates that K β L satellites do not significantly contribute to our measured spectra in the range where transient changes were observed. We therefore conclude that the effect of K β L satellites on the analysis and interpretation of our time resolved XES data is negligible. The overall similarity of our spectrum with the reported reference spectrum is consistent with the expected weak intensity of multivacancy satellites for strongly covalent compounds such as $^2[\text{Fe}^{\text{III}}(\text{CN})_6]^{3-}$ (Kawai, Nuclear Instruments and Methods in Physics Research B75 (1993) 3-8).

These findings have been added as Supplementary Note 8.

The intensity of the K-beta'' peak in the theoretical calculations for the ground state is significantly lower than in the experiment. It is unclear why and it adds some doubts regarding the method (or selected parameters) used to calculate the spectra. Is it possible to achieve better agreement of the intensities? How far are the relative intensities of Kbeta1,3 and Kbeta2,5 peaks in the theory and the experiment?

Multiple previous studies have reported a comparison of the experimental and calculated ground state spectra of $^2[\text{Fe}^{\text{III}}(\text{CN})_6]^{3-}$ and $^1[\text{Fe}^{\text{II}}(\text{CN})_6]^{4-}$. These studies have utilized ground state DFT or TDDFT with different combinations of exchange-correlation functionals and basis sets: Pollock and DeBeer (JACS 2011, 133, 5594-5601) / Lee *et al.* (JACS 2010, 132, 28, 9715-9727) have used the BP86 functional with TZVP and the expanded CP(PPP) basis set for Fe. Ross *et al.* (J. Phys. Chem. B 2018, 122, 19, 5075-5086) / Zhang *et al.* (J. Chem. Theory Comput. 2015, 11, 12, 5804-5809) have used the PBE0 functional combined with the Sapporo-TZP-2012 and 6-311G** basis sets for the Fe atom and light atoms, respectively. March *et al.* (J. Phys. Chem. C 2015, 119, 26, 14575-14578) have utilized the B3LYP/TZVP method.

Importantly, none of these approaches has accurately reproduced the relative intensities of the K β '' peak and the two peaks of the K $\beta_{2,5}$ feature.

More recently, we have also reported a comparison of the experimental and calculated $^2[\text{Fe}^{\text{III}}(\text{CN})_6]^{3-}$ and $^1[\text{Fe}^{\text{II}}(\text{CN})_6]^{4-}$ VtC spectrum using the B3LYP functional combined with the CP(PPP) and def2-TZVP basis sets for the Fe atoms and all other atoms, respectively (Ledbetter *et al.*, J. Chem. Phys. 152, 074203 (2020)). This study also shows the calculated dependence of relative VtC peak intensities on the metal-ligand distance.

Here, we have emphasized consistency with our previous study (Ledbetter *et al.*) and utilized the B3LYP functional combined with the CP(PPP) and ZORA-def2-TZVP basis sets for Fe and all other atoms, respectively.

As for the $K\beta_{1,3}$ and $K\beta_{2,5}$ peaks (experimentally $I_{\text{main}}/I_{\text{VtC}} \sim 20$), we have not compared the relative intensities of the calculated main line and VtC spectra, due to different theoretical approaches.

In the article Nature Chemistry 13, 343 (which has significant overlap in the list of co-authors with this manuscript) for the system with a similar structure around the Fe center the importance of solute-solvent interactions was demonstrated. Why this effect is negligible for $[\text{Fe}(\text{CN})_6]^{3-}$?

Biasin *et al.* (Nature Chemistry 13, 343-349 (2021)) reported the solvent response to the population of a short-lived metal-to-metal charge transfer excited state (~ 60 fs lifetime) in a mixed-valence bimetallic complex ($\text{Fe}^{\text{II}}\text{Ru}^{\text{III}}$) in water, photo-excited with 800 nm. The low-Q difference signal showed an oscillatory component within the first ~ 0.5 ps that was analyzed using non-equilibrium MD simulations. The signal was assigned to coherent translational motions of the first-solvation-shell water molecules arising from photoinduced changes in solute-solvent hydrogen bonding.

Here, for UV-excited aqueous $^2[\text{Fe}^{\text{III}}(\text{CN})_6]^{3-}$, we do observe a small negative difference signal in the low-Q range. However, this signal only starts growing after the decay of the $^2\text{T}_{2u}$ LMCT ES (~ 250 fs lifetime) and then decays in 4-5 ps. For these reasons, we have associated this difference signal with the penta-coordinate intermediate, rather than the $^2\text{T}_{2u}$ LMCT ES.

The absence of a significant structural response directly associated with the $^2\text{T}_{2u}$ LMCT ES is consistent with the modest intra-molecular structural rearrangement previously determined for the $^2\text{T}_{1u}$ LMCT ES. However, the reason for the absence of an experimentally observable structural response of the solvation shell to the population of the $^2\text{T}_{2u}$ LMCT ES remains unclear. While we agree that it is an important and intriguing question, it is not the focus of the present manuscript, and we believe it does not impact the validity of the conclusions drawn. Clarifying the absence of an observable structural response to the $^2\text{T}_{2u}$ LMCT ES will require further studies to better characterize the LMCT ES and a careful comparison with XSS signals measured for different photo-excited charge transfer ES. We would also like to emphasize that the

measurements presented in Biasin *et al.* differ in some crucial aspects from the measurements presented in this manuscript: Here, we utilize a significantly shorter excitation wavelength that should result in the rapid deposition of larger amounts of vibrational excess energy in the CN-bonds, which we speculate could impact the hydrogen bonding between the cyanides and the water molecules within tens of femtoseconds. We speculate that such effects could potentially diminish the response of the solvation shell to the charge transfer process, consistent with the lack of an observable XSS difference signal. Furthermore, the hydrogen bonding interactions of the complex may be sensitive to the σ - vs π -type MO occupancies on the cyanides and thus depend on the details of the charge transfer process.

Summarizing my overall impression from the manuscript: the experimental data I see as another good example of pump-probe measurements with an advanced LCLS instrument. Regarding the data interpretation, the authors are on a good track, but the analysis can be improved to make it convincing.

We thank the reviewer for acknowledging the experimental effort. Regarding the data interpretation, we have significantly extended the data analysis to further strengthen the critical points in the manuscript.

Reviewer 2:

The use of singular value decomposition (SVD) of the time resolved signal is an excellent approach to extract the dominant spectral changes and associated chemical species observed and mimics analysis in previous works, such as Ref 63.

We thank the reviewer for acknowledging the use of singular value decomposition shown in the supplementary information. We have further built on our SVD-based analysis by performing a global target analysis of the combined Fe K β main line and VtC spectral range. The resulting species associated difference spectra are now included in the manuscript in Figure 5. The details of this analysis are added as Supplementary Note 6.

There may be significant differences in the estimated relative energies of the various complexes when the first solvation shell of water molecules is included due to the presence of hydrogen-bonding between these molecules and the CN anions. Lack of H-bonds in the SP and TBP intermediates might raise their energies significantly. However, it is likely that the spectral simulations are not dramatically impacted by lack of inclusion of the solvating waters. The authors should comment on how much their analysis might differ if these details of solvation were considered.

This issue has been addressed in a previous study by Ross *et al.* (J. Phys. Chem. B 2018, 122, 5075-5086). The authors of this study have calculated the VtC spectrum of aqueous $^2[\text{Fe}^{\text{III}}(\text{CN})_6]^{3-}$ using TDDFT with both, explicit solute-solvent configurations and a dielectric continuum model (COSMO). The comparison of the $^2[\text{Fe}^{\text{III}}(\text{CN})_6]^{3-}$ VtC XES spectrum calculated with these two methods does not show significant differences. We therefore expect that our VtC XES simulations and related conclusions drawn in the manuscript are not significantly affected by neglecting the effects of explicit solvation.

Please provide an explanation of the spin-multiplicity notation for the various cyanide complexes - indicating that the first superscript number refers to the spin multiplicity (doublet, quartet, etc.) - not all previous publication have used this compact notation before and explaining it the first time it appears will benefit the general reader.

We have added a brief clarification in the introduction where the notation appears for the first time.

Reviewer 3:

The authors use a combination of post-Hartree Fock theory (RASSCF + CASPT2) for the K β and density functional theory for the valence to core emission. Why not using RASSCF for both? I assume that the size of the active space becomes impractical, but could the limitations of DFT + one electron approach for VtC transitions bring some issues?

The VtC XES calculation using RASSCF + CASPT2 requires a much larger active space by including extra sets of occupied valence orbitals, moreover the VtC XES calculation also needs to calculate a lot of final states to reproduce the spectral features, for the current systems without evident symmetry, the computational cost of using RASSCF+CASPT2 for VtC XES becomes impractical. The DFT approach is a good alternative to calculate the VtC XES and it has been successfully applied to calculate VtC XES of transition metal complexes (see Lee *et al.* JACS 2010, 132 (28), 9715-9727). The one electron approach has limitation to capture peaks with multi-electron excitation character, which have little contribution to the intensity of VtC XES.

Does the different level of theory used for different part of the spectra complicates the analysis and merging of spectra?

The calculations of K β main line XES and VtC XES are independent from each other by using different theory levels, but they all support the same conclusion made in this project. From the theoretical interpretation perspective, we believe that consistent conclusions drawn from independent calculations are more reliable and convincing.

It would be convenient to write explicitly in SI the fitting equations used for different models (not necessarily all of them) to compare with the summarizing tables at a glance. Now they can be deduced from the text.

We have added the general fit equation (Equation 1) to Supplementary Note 3 and now refer to it throughout the text.

Some comments could be added on how error bars are computed, e.g. in Fig 2a and b (standard deviation over some number of shots, propagated uncertainty, enlarged or not to account for a confidence interval, etc). Similarly, uncertainties of the fitted parameters could be described more (are the datapoints weighted by their experimental uncertainty in the fitting routine, how are calculated the uncertainties of the parameters of the nonlinear fitting model, etc).

For Figures 2a-c, Figure 3a, and Figure 4b we have added an explanation of the uncertainties in the figure caption. For Figures 3b-c, we now refer to Supplementary Note 4 in the figure caption, where the procedure used to estimate the uncertainties is described.

the article concludes with an opening toward general transition metal sites in complex environments. The authors could comment on the applicability of the technique to study proteins, for example small molecule dissociation in heme proteins or other systems that interest them.

We have added the following sentence to the end of the manuscript: “Emerging capabilities at x-ray free electron lasers could for instance enable deeper studies of photoinduced active site dynamics in more dilute, biologically relevant systems such as heme proteins.”

Reviewer #1 (Remarks to the Author):

The authors adequately revised the manuscript by responding to my questions and doubts about the data analysis. Not so much has been done to respond to the criticism about relative intensities of features in the calculations of valence-to-core XES (apart from the extensive list of references in the rebuttal letter with the statement that "none of these approaches has accurately reproduced the relative intensities of the $K\beta'$ peak and the two peaks of the $K\beta_{2,5}$ feature"). Nevertheless, this aspect does not influence the main conclusions of the manuscript. Therefore, I think the manuscript can be published in its current form.

Reviewer #2 (Remarks to the Author):

My concerns during the first review have been addressed adequately by the authors. No further review required.

Reviewer #3 (Remarks to the Author):

The authors have addressed all the comments properly.
The manuscript is worthy of publication in Nature Communications.